# Non-Stationary Markov Decision Processes a Worst-Case Approach using Model-Based Reinforcement Learning

**Erwan Lecarpentier**
Université de Toulouse
ONERA - The French Aerospace Lab
erwan.lecarpentier@isae-supaero.fr

**Emmanuel Rachelson**
Université de Toulouse
ISAE-SUPAERO
emmanuel.rachelson@isae-supaero.fr

## Abstract

This work tackles the problem of robust planning in non-stationary stochastic environments. We study Markov Decision Processes (MDPs) evolving over time and consider Model-Based Reinforcement Learning algorithms in this setting. We make two hypotheses: 1) the environment evolves continuously with a bounded evolution rate; 2) a current model is known at each decision epoch but not its evolution. Our contribution can be presented in four points. 1) we define a specific class of MDPs that we call Non-Stationary MDPs (NSMDPs). We introduce the notion of regular evolution by making an hypothesis of Lipschitz-Continuity on the transition and reward functions w.r.t. time; 2) we consider a planning agent using the current model of the environment but unaware of its future evolution. This leads us to consider a worst-case method where the environment is seen as an adversarial agent; 3) following this approach, we propose the Risk-Averse Tree-Search (RATS) algorithm, a Model-Based method similar to minimax search; 4) we illustrate the benefits brought by RATS empirically and compare its performance with reference Model-Based algorithms.

## 1 Introduction

One of the hot topics of modern Artificial Intelligence (AI) is the ability for an agent to adapt its behavior to changing tasks. In the literature, this problem is often linked to the setting of Lifelong Reinforcement Learning (LRL) [Silver et al., 2013, Abel et al., 2018a,b] and learning in non-stationary environments [Choi et al., 1999, Jaulmes et al., 2005, Hadoux, 2015]. In LRL, the tasks presented to the agent change sequentially at discrete transition epochs [Silver et al., 2013]. Similarly, the non-stationary environments considered in the literature often evolve abruptly [Hadoux, 2015, Hadoux et al., 2014, Doya et al., 2002, Da Silva et al., 2006, Choi et al., 1999, 2000, 2001, Campo et al., 1991, Wiering, 2001]. In this paper, we investigate environments continuously changing over time that we call Non-Stationary Markov Decision Processes (NSMDPs). In this setting, it is realistic to bound the evolution rate of the environment using a Lipschitz Continuity (LC) assumption.

Model-based Reinforcement Learning approaches [Sutton et al., 1998] benefit from the knowledge of a model allowing them to reach impressive performances, as demonstrated by the Monte Carlo Tree Search (MCTS) algorithm [Silver et al., 2016]. In this matter, the necessity to have access to a model is a great concern of AI [Asadi et al., 2018, Jaulmes et al., 2005, Doya et al., 2002, Da Silva et al., 2006]. In the context of NSMDPs, we assume that an agent is provided with a *snapshot* model when its action is computed. By this, we mean that it only has access to the current model of the environment but not its future evolution, as if it took a photograph but would be unable to predict how it is going to evolve. This hypothesis is realistic, because many environments have a tractable state while their future evolution is hard to predict [Da Silva et al., 2006, Wiering, 2001]. In order to solve

LC-NSMDPs, we propose a method that considers the worst-case possible evolution of the model and performs planning w.r.t. this model. This is equivalent to considering Nature as an adversarial agent. The paper is organized as follows: first we describe the NSMDP setting and the regularity assumption (Section 2); then we outline related works (Section 3); follows the explanation of the worst-case approach proposed in this paper (Section 4); then we describe an algorithm reflecting this approach (Section 5); finally we illustrate its behavior empirically (Section 6).

## 2   Non-Stationary Markov Decision Processes

To define a Non-Stationary Markov Decision Process (NSMDP), we revert to the initial MDP model introduced by Puterman [2014], where the transition and reward functions depend on time.

**Definition 1.** *NSMDP. An NSMDP is an MDP whose transition and reward functions depend on the decision epoch. It is defined by a 5-tuple $\{S, T, A, (p_t)_{t \in T}, (r_t)_{t \in T}\}$ where $S$ is a state space; $T \equiv \{1, 2, \ldots, N\}$ is the set of decision epochs with $N \leq +\infty$; $A$ is an action space; $p_t(s' \mid s, a)$ is the probability of reaching state $s'$ while performing action $a$ at decision epoch $t$ in state $s$; $r_t(s, a, s')$ is the scalar reward associated to the transition from $s$ to $s'$ with action $a$ at decision epoch $t$.*

This definition can be viewed as that of a stationary MDP whose state space has been enhanced with time. While this addition is trivial in episodic tasks where an agent is given the opportunity to interact several times with the same MDP, it is different when the experience is unique. Indeed, no exploration is allowed along the temporal axis. Within a stationary, infinite-horizon MDP with a discounted criterion, it is proven that there exists a Markovian deterministic stationary policy [Puterman, 2014]. It is not the case within NSMDPs where the optimal policy is non-stationary in the most general case. Additionally, we define the expected reward received when taking action $a$ at state $s$ and decision epoch $t$ as $R_t(s, a) = \mathbb{E}_{s' \sim p_t(\cdot \mid s, a)} [r_t(s, a, s')]$. Without loss of generality, we assume the reward function to be bounded between $-1$ and $1$. In this paper, we consider discrete time decision processes with constant transition durations, which imply deterministic decision times in Definition 1. This assumption is mild since many discrete time sequential decision problems follow that assumption. A non-stationary policy $\pi$ is a sequence of *decision rules* $\pi_t$ which map states to actions (or distributions over actions). For a stochastic non-stationary policy $\pi_t(a \mid s)$, the value of a state $s$ at decision epoch $t$ within an infinite horizon NSMDP is defined, with $\gamma \in [0, 1)$ a discount factor, by:

$$V_t^\pi(s) = \mathbb{E}\left[\sum_{i=t}^{\infty} \gamma^{i-t} R_i(s_i, a_i) \,\middle|\, s_t = s, \ a_i \sim \pi_i(\cdot \mid s_i), \ s_{i+1} \sim p_i(\cdot \mid s_i, a_i) \right],$$

The definition of the state-action value function $Q_t^\pi$ for $\pi$ at decision epoch $t$ is straightforward:

$$Q_t^\pi(s, a) = R_t(s, a) + \gamma \mathop{\mathbb{E}}_{s' \sim p_t(\cdot \mid s, a)} \left[V_{t+1}^\pi(s')\right].$$

Overall, we defined an NSMDP as an MDP where we stress out the distinction between state, time, and decision epoch due to the inability for an agent to explore the temporal axis at will. This distinction is particularly relevant for non-episodic tasks, i.e. when there is no possibility to re-experience the same MDP starting from a prior date.

**The regularity hypothesis.** Many real-world problems can be modeled as an NSMDP. For instance, the problem of path planning for a glider immersed in a non-stationary atmosphere [Chung et al., 2015, Lecarpentier et al., 2017], or that of vehicle routing in dynamic traffic congestion. Realistically, we consider that the expected reward and transition functions do not evolve arbitrarily fast over time. Conversely, if such an assumption was not made, a chaotic evolution of the NSMDP would be allowed which is both unrealistic and hard to solve. Hence, we assume that changes occur slowly over time. Mathematically, we formalize this hypothesis by bounding the evolution rate of the transition and expected reward functions, using the notion of Lipschitz Continuity (LC).

**Definition 2.** *Lipschitz Continuity. Let $(X, d_X)$ and $(Y, d_Y)$ be two metric spaces and $f : X \to Y$, $f$ is L-Lipschitz Continuous (L-LC) with $L \in \mathbb{R}^+$ iff $d_Y(f(x), f(\hat{x})) \leq L \, d_X(x, \hat{x}), \forall(x, \hat{x}) \in X^2$. $L$ is called a Lipschitz constant of the function $f$.*

We apply this hypothesis to the transition and reward functions of an NSMDP so that those functions are LC w.r.t. time. For the transition function, this leads to the consideration of a metric between probability density functions. For that purpose, we use the 1-Wasserstein distance [Villani, 2008].

**Definition 3. *1-Wasserstein distance.*** *Let $(X, d_X)$ be a Polish metric space, $\mu, \nu$ any probability measures on $X$, $\Pi(\mu, \nu)$ the set of joint distributions on $X \times X$ with marginals $\mu$ and $\nu$. The 1-Wasserstein distance between $\mu$ and $\nu$ is $W_1(\mu, \nu) = \inf_{\pi \in \Pi(\mu, \nu)} \int_{X \times X} d_X(x, y) d\pi(x, y)$.*

The choice of the Wasserstein distance is motivated by the fact that it quantifies the distance between two distributions in a physical manner, respectful of the topology of the measured space [Dabney et al., 2018, Asadi et al., 2018]. First, it is sensitive to the difference between the supports of the distributions. Comparatively, the Kullback-Leibler divergence between distributions with disjoint supports is infinite. Secondly, if one consider two regions of the support where two distributions differ, the Wasserstein distance is sensitive to the distance between the elements of those regions. Comparatively, the total-variation metric is the same regardless of this distance.

**Definition 4.** $(L_p, L_r)$**-LC-NSMDP.** *An $(L_p, L_r)$-LC-NSMDP is an NSMDP whose transition and reward functions are respectively $L_p$-LC and $L_r$-LC w.r.t. time, i.e., $\forall (t, \hat{t}, s, s', a) \in \mathcal{T}^2 \times \mathcal{S}^2 \times \mathcal{A}$,*

$$W_1(p_t(\cdot \mid s, a), p_{\hat{t}}(\cdot \mid s, a)) \leq L_p |t - \hat{t}| \quad and \quad |r_t(s, a, s') - r_{\hat{t}}(s, a, s')| \leq L_r |t - \hat{t}|.$$

One should remark that the LC property should be defined with respect to actual decision times and not decision epoch indexes for the sake of realism. In the present case, both have the same value, and we choose to keep this convention for clarity. Our results however extend easily to the case where indexes and times do not coincide. From now on, we consider $(L_p, L_r)$-LC-NSMDPs, making Lipschitz Continuity our regularity property. Notice that $R$ is defined as a convex combination of $r$ by the probability measure $p$. As a result, the notion of Lipschitz Continuity of $R$ is strongly related to that of $r$ and $p$ as showed by Property 1. All the proofs of the paper can be found in the Appendix.

**Property 1.** *Given an $(L_p, L_r)$-LC-NSMDP, the expected reward function $R_t : s, a \mapsto \mathbb{E}_{s' \sim p_t(\cdot|s,a)} \{r_t(s, a, s')\}$ is $L_R$-LC with $L_R = L_r + L_p$.*

This result shows $R$'s evolution rate is conditioned by the evolution rates of $r$ and $p$. It allows to work either with the reward function $r$ or its expectation $R$, benefiting from the same LC property.

## 3 Related work

Iyengar [2005] introduced the framework of robust MDPs, where the transition function is allowed to evolve within a set of functions due to uncertainty. This differs from our work in two fundamental aspects: 1) we consider uncertainty in the reward model as well; 2) we use a stronger Lipschitz formulation on the set of possible transition and reward functions, this last point being motivated by its relevance to the non-stationary setting. Szita et al. [2002] also consider the robust MDP setting and adopt a different constraint hypothesis on the set of possible functions than our LC assumption. They control the total variation distance of transition functions from subsequent decision epochs by a scalar value. Those slowly changing environments allow model-free RL algorithms such as Q-Learning to find near optimal policies. Lim et al. [2013] consider learning in robust MDPs where the model evolves in an adversarial manner for a subset of $\mathcal{S} \times \mathcal{A}$. In that setting, they propose to learn to what extent the adversary can modify the model and to deduce a behavior close to the minimax policy. Even-Dar et al. [2009] studied the case of non-stationary reward functions with fixed transition models. No assumption is made on the set of possible functions and they propose an algorithm achieving sub-linear regret w.r.t. the best stationary policy. Dick et al. [2014] viewed a similar setting from the perspective of online linear optimization. Csáji and Monostori [2008] studied the NSMDP setting with an assumption of reward and transition functions varying in a neighborhood of a reference reward-transition function pair. Finally, Abbasi et al. [2013] address the adversarial NSMDP setting with a mixing assumption constraint instead of the LC assumption we make.

Non-stationary environments also have been studied through the framework of Hidden Mode MDPs (HM-MDP) introduced by Choi et al. [1999]. This is a special class of Partially Observable MDPs (POMDPs) [Kaelbling et al., 1998] where a hidden mode indexes a latent stationary MDP within which the agent evolves. Similarly to the context of LRL, the agent experiences a series of different MDPs over time. In this setting, Choi et al. [1999, 2000] proposed methods to learn the different models of the latent stationary MDPs. Doya et al. [2002] built a modular architecture switching between models and policies when a change is detected. Similarly, Wiering [2001], Da Silva et al. [2006], Hadoux et al. [2014] proposed a method tracking the switching occurrence and re-planning if needed. Overall, as in LRL, the HM-MDP setting considers abrupt evolution of the transition and

reward functions whereas we consider a continuous one. Other settings have been considered, as by Jaulmes et al. [2005], who do not make particular hypothesis on the evolution of the NSMDP. They build a learning algorithm for POMDPs solving, weighting recently experienced transitions more than older ones to account for the time dependency.

To plan robustly within an NSMDP, our approach consists in exploiting the *slow* LC evolution of the environment. Utilizing Lipschitz continuity to infer bounds on a function is common in the RL, bandit and optimization communities [Kleinberg et al., 2008, Rachelson and Lagoudakis, 2010, Pirotta et al., 2015, Pazis and Parr, 2013, Munos, 2014]. We implement this approach with a minimax-like algorithm [Fudenberg and Tirole, 1991], where the environment is seen as an adversarial agent.

## 4 Worst-case approach

We consider finding an optimal policy within an LC-NSMDP under the non-episodic task hypothesis. The latter prevents us from learning from previous experience data since they become outdated with time and no information samples have been collected yet for future time steps. An alternative is to use model-based RL algorithms such as MCTS. For a current state $s_0$, such algorithms focus on finding the optimal action $a_0^*$ by using a generative model. This action is then undertaken and the operation repeated at the next state. However, using the true NSMDP model for this purpose is an unrealistic hypothesis, since this model is generally unknown. We assume the agent does not have access to the true NSMDP model; instead, we introduce the notion of *snapshot model*. Intuitively, the snapshot associated to time $t_0$ is a temporal slice of the NSMDP at $t_0$.

**Definition 5.** *Snapshot of an NSMDP. The snapshot of an NSMDP $\{\mathcal{S}, \mathcal{T}, \mathcal{A}, (p_t)_{t \in \mathcal{T}}, (r_t)_{t \in \mathcal{T}}\}$ at decision epoch $t_0$, denoted by $\mathrm{MDP}_{t_0}$, is the stationary MDP defined by the 4-tuple $\{\mathcal{S}, \mathcal{A}, p_{t_0}, r_{t_0}\}$ where $p_{t_0}(s' \mid s, a)$ and $r_{t_0}(s, a, s')$ are the transition and reward functions of the NSMDP at $t_0$.*

Similarly to the NSMDP, this definition induces the existence of the snapshot expected reward $R_{t_0}$ defined by $R_{t_0} : s, a \mapsto \mathbb{E}_{s' \sim p_{t_0}(\cdot \mid s, a)} \{r_{t_0}(s, a, s')\}$. Notice that the snapshot $\mathrm{MDP}_{t_0}$ is stationary and coincides with the NSMDP *only* at $t_0$. Particularly, one can generate a trajectory $\{s_0, r_0, \cdots, s_k\}$ within an NSMDP using the sequence of snapshots $\{\mathrm{MDP}_{t_0}, \cdots, \mathrm{MDP}_{t_0+k-1}\}$ as a model. Overall, the hypothesis of using snapshot models amounts to considering a planning agent only able to get the current stationary model of the environment. In real-world problems, predictions often are uncertain or hard to perform e.g. in the thermal soaring problem of a glider.

We consider a generic *planning* agent at $s_0, t_0$, using $\mathrm{MDP}_{t_0}$ as a model of the NSMDP. By planning, we mean conducting a look-ahead search within the possible trajectories starting from $s_0, t_0$ given a model of the environment. The search allows in turn to identify an optimal action w.r.t. the model. This action is then undertaken and the agent jumps to the next state where the operation is repeated. The consequence of planning with $\mathrm{MDP}_{t_0}$ is that the estimated value of an $s, t$ pair is the value of the optimal policy of $\mathrm{MDP}_{t_0}$, written $V^*_{\mathrm{MDP}_{t_0}}(s)$. The true optimal value of $s$ at $t$ within the NSMDP does not match this estimate because of the non-stationarity. The intuition we develop is that, given the *slow* evolution rate of the environment, for a state $s$ seen at a future decision epoch during the search, we can predict a scope into which the transition and reward functions at $s$ lie.

**Property 2.** *Set of admissible snapshot models. Consider an $(L_p, L_r)$-LC-NSMDP, $s, t, a \in \mathcal{S} \times \mathcal{T} \times \mathcal{A}$. The transition and expected reward functions $(p_t, R_t)$ of the snapshot $\mathrm{MDP}_t$ respect*

$$(p_t, R_t) \in \Delta_t := \mathcal{B}_{W_1}(p_{t-1}(\cdot \mid s, a), L_p) \times \mathcal{B}_{|\cdot|}(R_{t-1}(s, a), L_R)$$

*where $L_R = L_p + L_r$ and $\mathcal{B}_d(c, r)$ denotes the ball of centre $c$, defined with metric $d$ and radius $r$.*

For a future prediction at $s, t$, we consider the question of using a better model than $p_{t_0}, R_{t_0}$. The underlying evolution of the NSMDP being unknown, a desirable feature would be to use a model leading to a policy that is *robust* to every possible evolution. To that end, we propose to use the snapshots corresponding to the worst possible evolution scenario under the constraints of Property 2. We claim that such a practice is an efficient way to 1) ensure robust performance to all possible evolutions of the NSMDP and 2) avoid catastrophic terminal states. Practically, this boils down to using a different value estimate for $s$ at $t$ than $V^*_{\mathrm{MDP}_{t_0}}(s)$ which provided no robustness guarantees.

Given a policy $\pi = (\pi_t)_{t \in \mathcal{T}}$ and a decision epoch $t$, a worst-case NSMDP corresponds to a sequence of transition and reward models minimizing the expected value of applying $\pi$ in any pair $(s, t)$, while

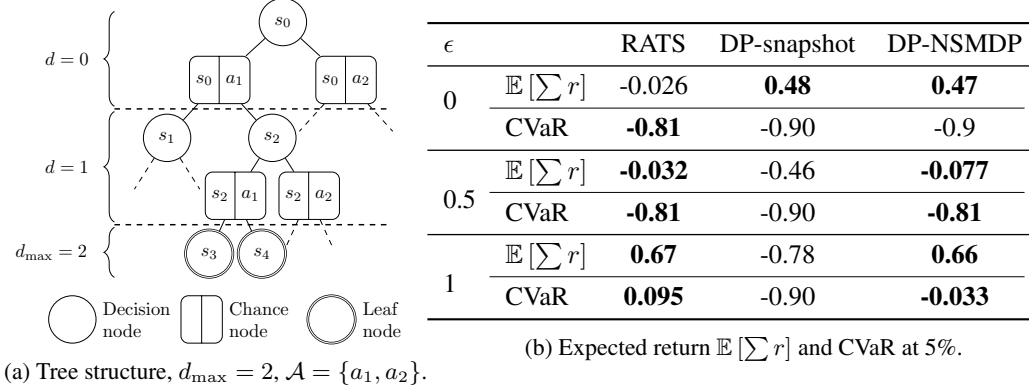

(a) Tree structure, $d_{\max} = 2$, $\mathcal{A} = \{a_1, a_2\}$.

| $\epsilon$ | | RATS | DP-snapshot | DP-NSMDP |
|---|---|---|---|---|
| 0 | $\mathbb{E}\left[\sum r\right]$ | -0.026 | **0.48** | **0.47** |
| | CVaR | **-0.81** | -0.90 | -0.9 |
| 0.5 | $\mathbb{E}\left[\sum r\right]$ | **-0.032** | -0.46 | **-0.077** |
| | CVaR | **-0.81** | -0.90 | **-0.81** |
| 1 | $\mathbb{E}\left[\sum r\right]$ | **0.67** | -0.78 | **0.66** |
| | CVaR | **0.095** | -0.90 | **-0.033** |

(b) Expected return $\mathbb{E}\left[\sum r\right]$ and CVaR at 5%.

Figure 1: Tree structure and results from the Non-Stationary bridge experiment.

remaining within the bounds of Property 2. We write $\overline{V}_t^\pi(s)$ this value for $s$ at decision epoch $t$.

$$\overline{V}_t^\pi(s) := \min_{(p_i, R_i) \in \Delta_i, \forall i \in \mathcal{T}} \mathbb{E}\left[\sum_{i=t}^\infty \gamma^{i-t} R_i(s_i, a_i) \,\bigg|\, \begin{matrix} s_t = s \\ a_i \sim \pi_i(\cdot \mid s_i), s_{i+1} \sim p_i(\cdot \mid s_i, a_i) \end{matrix}\right] \quad (1)$$

Intuitively, the worst-case NSMDP is a model of a non-stationary environment leading to the poorest possible performance for $\pi$, while being an admissible evolution of MDP$_t$. Let us define $\overline{Q}_t^\pi(s, a)$ as the worst-case $Q$-value for the pair $(s, a)$ at decision epoch $t$:

$$\overline{Q}_t^\pi(s, a) := \min_{(p, R) \in \Delta_t} \mathbb{E}_{s' \sim p}\left[R(s, a) + \gamma \overline{V}_{t+1}^\pi(s')\right]. \quad (2)$$

## 5 Risk-Averse Tree-Search algorithm

**The algorithm.** Tree search algorithms within MDPs have been well studied and cover two classes of search trees, namely closed loop [Keller and Helmert, 2013, Kocsis and Szepesvári, 2006, Browne et al., 2012] and open loop [Bubeck and Munos, 2010, Lecarpentier et al., 2018]. Following [Keller and Helmert, 2013], we consider closed loop search trees, composed of *decision nodes* alternating with *chance nodes*. We adapt their formulation to take time into account, resulting in the following definitions. A decision node at depth $t$, denoted by $\nu^{s,t}$, is labeled by a unique state / decision epoch pair $(s, t)$. The edges leading to its children chance nodes correspond to the available actions at $(s, t)$. A chance node, denoted by $\nu^{s,t,a}$, is labeled by a state / decision epoch / action triplet $(s, t, a)$. The edges leading to its children decision nodes correspond to the reachable state / decision epoch pairs $(s', t')$ after performing $a$ in $(s, t)$ as illustrated by Figure 1a. We consider the problem of estimating the optimal action $a_0^*$ at $s_0, t_0$ within a worst-case NSMDP, knowing MDP$_{t_0}$. This problem is twofold. It requires 1) to estimate the worst-case NSMDP given MDP$_{t_0}$ and 2) to explore the latter in order to identify $a_0^*$. We propose to tackle both problems with an algorithm inspired by the minimax algorithm [Fudenberg and Tirole, 1991] where the *max* operator corresponds to the agent's policy, seeking to maximize the return; and the *min* operator corresponds to the worst-case model, seeking to minimize the return. Estimating the worst-case NSMDP requires to estimate the sequence of subsequent snapshots minimizing Equation 2. The inter-dependence of those snapshots (Equation 1) makes the problem hard to solve [Iyengar, 2005], particularly because of the combinatorial nature of the opponent's action space. Instead, we propose to solve a relaxation of this problem, by considering snapshots only constrained by MDP$_{t_0}$. Making this approximation leaves a possibility to violate property 2 but allows for an efficient search within the developed tree and (as will be shown experimentally) leads to robust policies. For that purpose, we define the set of admissible snapshot models w.r.t. MDP$_{t_0}$ by $\Delta_{t_0}^t := \mathcal{B}_{W_1}\left(p_{t_0}(\cdot \mid s, a), L_p|t - t_0|\right) \times \mathcal{B}_{|\cdot|}\left(R_{t_0}(s, a), L_R|t - t_0|\right)$. The relaxed analogues of Equations 1 and 2 for $s, t, a \in \mathcal{S} \times \mathcal{T} \times \mathcal{A}$ are defined as follows:

$$\hat{V}_{t_0, t}^\pi(s) := \min_{(p_i, R_i) \in \Delta_{t_0}^i, \forall i \in \mathcal{T}} \mathbb{E}\left[\sum_{i=t}^\infty \gamma^{i-t} R_i(s_i, a_i) \,\bigg|\, \begin{matrix} s_t = s \\ a_i \sim \pi_i(\cdot \mid s_i), s_{i+1} \sim p_i(\cdot \mid s_i, a_i) \end{matrix}\right],$$

$$\hat{Q}_{t_0, t}^\pi(s, a) := \min_{(p, R) \in \Delta_{t_0}^t} \mathbb{E}_{s' \sim p}\left[R(s, a) + \gamma \hat{V}_{t_0, t+1}^\pi(s')\right].$$

---

**Algorithm 1:** RATS algorithm

---

**RATS** ($s_0$, $t_0$, maxDepth)
$\nu_0$ = rootNode($s_0$, $t_0$)
Minimax($\nu_0$)
$\nu^* = \arg\max_{\nu' \text{ in } \nu_0.\text{children}} \nu'.\text{value}$
**return** $\nu^*$.action

**Minimax** ($\nu$, maxDepth)
**if** $\nu$ *is DecisionNode* **then**
    **if** $\nu$.state *is terminal* **or** $\nu$.depth = maxDepth **then**
        **return** $\nu$.value = heuristicValue($\nu$.state)
    **else**
        **return** $\nu$.value = $\max_{\nu' \in \nu.\text{children}}$Minimax($\nu'$, maxDepth)
**else**
    **return** $\nu$.value = $\min_{(p,R)\in\Delta_{t_0}^t} R(\nu) + \gamma \sum_{\nu'\in\nu.\text{children}} p(\nu' \mid \nu)$Minimax($\nu'$, maxDepth)

---

Their optimal counterparts, while seeking to find the optimal policy, verify the following equations:

$$\hat{V}_{t_0,t}^*(s) = \max_{a\in\mathcal{A}} \hat{Q}_{t_0,t}^*(s,a), \tag{3}$$

$$\hat{Q}_{t_0,t}^*(s,a) = \min_{(p,R)\in\Delta_{t_0}^t} \mathop{\mathbb{E}}_{s'\sim p} \left[ R(s,a) + \gamma \hat{V}_{t_0,t+1}^*(s') \right]. \tag{4}$$

We now provide a method to calculate those quantities within the nodes of the tree search algorithm. **Max nodes.** A decision node $\nu^{s,t}$ corresponds to a max node due to the greediness of the agent w.r.t. the subsequent values of the children. We aim at maximizing the return while retaining a risk-averse behavior. As a result, the value of $\nu^{s,t}$ follows Equation 3 and is defined as:

$$V(\nu^{s,t}) = \max_{a\in\mathcal{A}} V(\nu^{s,t,a}). \tag{5}$$

**Min nodes.** A chance node $\nu^{s,t,a}$ corresponds to a min node due to the use of a worst-case NSMDP as a model which minimizes the value of $\nu^{s,t,a}$ w.r.t. the reward and the subsequent values of its children. Writing the value of $\nu^{s,t,a}$ as the value of $s, t, a$, within the worst-case snapshot minimizing Equation 4, and using the children's values as values for the next reachable states, leads to Equation 6.

$$V(\nu^{s,t,a}) = \min_{(p,R)\in\Delta_{t_0}^t} R(s,a) + \gamma \mathop{\mathbb{E}}_{s'\sim p} V(\nu^{s',t+1}) \tag{6}$$

Our approach considers the environment as an adversarial agent, as in an *asymmetric* two-player game, in order to search for a robust plan. The resulting algorithm, RATS for Risk-Averse Tree-Search, is described in Algorithm 1. Given an initial state / decision epoch pair, a minimax tree is built using the snapshot MDP$_{t_0}$ and the operators corresponding to Equations 5 and 6 in order to estimate the worst-case snapshots at each depth. The tree is built, the action leading to the best possible value from the root node is selected and a real transition is performed. The next state is then reached, the new snapshot model MDP$_{t_0+1}$ is acquired and the process re-starts. Notice the use of $R(\nu)$ and $p(\nu' \mid \nu)$ in the pseudo-code: they are light notations respectively standing for $R_t(s,a)$ corresponding to a chance node $\nu \equiv \nu^{s,t,a}$ and the probability $p_t(s'|s,a)$ to jump to a decision node $\nu' \equiv \nu^{s',t+1}$ given a chance node $\nu \equiv \nu^{s,t,a}$. The tree built by RATS is entirely developed until the maximum depth $d_{\max}$. A heuristic function is used to evaluate the leaf nodes of the tree.

**Analysis of RATS.** We are interested in characterizing Algorithm 1 without function approximation and therefore will consider finite, countable, $\mathcal{S} \times \mathcal{A}$ sets. We now detail the computation of the min operator (Property 3), the computational complexity of RATS (Property 4) and the heuristic function.

**Property 3.** ***Closed-form expression of the worst case snapshot of a chance node.*** *Following Algorithm 1, a solution to Equation 6 is given by:*

$$\hat{R}(s,a) = R_{t_0}(s,a) - L_R|t-t_0| \quad and \quad \hat{p}(\cdot \mid s,a) = (1-\lambda)p_{t_0}(\cdot \mid s,a) + \lambda p_{sat}(\cdot \mid s,a)$$

*with* $p_{sat}(\cdot \mid s,a) = (0,\cdots,0,1,0,\cdots,0)$ *with* 1 *at position* $\arg\min_{s'} V(\nu^{s',t+1})$, $\lambda = 1$ *if* $W_1(p_{sat},p_0) \le L_p|t-t_0|$ *and* $\lambda = L_p|t-t_0|/W_1(p_{sat},p_0)$ *otherwise.*

**Property 4.** *Computational complexity. The total computation complexity of Algorithm 1 is* $\mathcal{O}(B|\mathcal{S}|^{1.5}|\mathcal{A}|\,(|\mathcal{S}||\mathcal{A}|)^{d_{\max}})$ *with $B$ the number of time steps and $d_{\max}$ the maximum depth.*

**Heuristic function.** As in vanilla minimax algorithms, Algorithm 1 bootstraps the values of the leaf nodes with a heuristic function if these leaves do not correspond to terminal states. Given such a leaf node $\nu^{s,t}$, a heuristic aims at estimating the value of the optimal policy at $(s,t)$ within the worst-case NSMDP, i.e. $\hat{V}^*_{t_0,t}(s)$. Let $H(s,t)$ be such a heuristic function, we call *heuristic error* in $(s,t)$ the difference between $H(s,t)$ and $\hat{V}^*_{t_0,t}(s)$. Assuming that the heuristic error is uniformly bounded, the following property provides an upper bound on the propagated error due to the choice of $H$.

**Property 5.** *Upper bound on the propagated heuristic error within RATS. Consider an agent executing Algorithm 1 at $s_0, t_0$ with a heuristic function $H$. We note $\mathcal{L}$ the set of all leaf nodes. Suppose that the heuristic error is uniformly bounded, i.e. $\exists \delta > 0, \forall \nu^{s,t} \in \mathcal{L}, |H(s) - \hat{V}^*_{t_0,t}(s)| \leq \delta$. Then we have for every decision and chance nodes $\nu^{s,t}$ and $\nu^{s,t,a}$, at any depth $d \in [0, d_{\max}]$:*

$$|V(\nu^{s,t}) - \hat{V}^*_{t_0,t}(s)| \leq \gamma^{(d_{\max}-d)}\delta \quad \text{and} \quad |V(\nu^{s,t,a}) - \hat{Q}^*_{t_0,t}(s,a)| \leq \gamma^{(d_{\max}-d)}\delta.$$

This last result implies that with *any* heuristic function $H$ inducing a uniform heuristic error, the propagated error at the root of the tree is guaranteed to be upper bounded by $\gamma^{d_{\max}}\delta$. In particular, since the reward function is bounded by hypothesis, we have $\hat{V}^*_{t_0,t}(s) \leq 1/(1-\gamma)$. Thus, selecting for instance the zero function ensures a root node heuristic error of at most $\gamma^{d_{\max}}/(1-\gamma)$. In order to improve the precision of the algorithm, we propose to guide the heuristic by using a function reflecting better the value of state $s$ at leaf node $\nu^{s,t}$. The ideal function would of course be $H(s) = \hat{V}^*_{t_0,t}(s)$, reducing the heuristic error to zero, but this is intractable. Instead, we suggest to use the value of $s$ within the snapshot $MDP_t$ using an evaluation policy $\pi$, i.e. $H(s) = V^\pi_{\mathrm{MDP}_t}(s)$. This snapshot is also not available, but Property 6 provides a range wherein this value lies.

**Property 6.** *Bounds on the snapshots values. Let $s \in \mathcal{S}$, $\pi$ a stationary policy, $MDP_{t_0}$ and $MDP_t$ two snapshot MDPs, $t, t_0 \in \mathcal{T}^2$ be. We note $V^\pi_{MDP_i}(s)$ the value of $s$ within $MDP_i$ following $\pi$. Then,*

$$|V^\pi_{MDP_{t_0}}(s) - V^\pi_{MDP_t}(s)| \leq |t - t_0|L_R/(1-\gamma).$$

Since $MDP_{t_0}$ is available, $V^\pi_{\mathrm{MDP}_{t_0}}(s)$ can be estimated, e.g. via Monte-Carlo roll-outs. Let $\widehat{V}^\pi_{\mathrm{MDP}_{t_0}}(s)$ denote such an estimate. Following Property 6, $V^\pi_{\mathrm{MDP}_{t_0}}(s) - |t - t_0|L_R/(1-\gamma) \leq V^\pi_{\mathrm{MDP}_t}(s)$. Hence, a worst-case heuristic on $V^\pi_{\mathrm{MDP}_t}(s)$ is $H(s) = \widehat{V}^\pi_{\mathrm{MDP}_{t_0}}(s) - |t - t_0|L_R/(1-\gamma)$. The bounds provided by Property 5 decrease quickly with $d_{\max}$, and given that $d_{\max}$ is *large enough*, RATS provides the optimal risk-averse maximizing the worst-case value for any evolution of the NSMDP.

## 6  Experiments

We compare the RATS algorithm with two policies [1]. The first one, named DP-snapshot, uses Dynamic Programming to compute the optimal actions w.r.t. the snapshot models at each decision epoch. The second one, named DP-NSMDP, uses the real NSMDP as a model to provide its optimal action. The latter behaves as an omniscient agent and should be seen as an upper bound on the performance. We choose a particular grid-world domain coined "Non-Stationary bridge" illustrated in Appendix, Section 7. An agent starts at the state labeled S in the center and the goal is to reach one of the two terminal states labeled G where a reward of +1 is received. The gray cells represent holes that are terminal states where a reward of -1 is received. Reaching the goal on the right leads to the highest payoff since it is closest to the initial state and a discount factor $\gamma = 0.9$ is applied. The actions are $\mathcal{A} = \{\text{Up, Right, Down, Left}\}$. The transition function is stochastic and non-stationary. At decision epoch $t = 0$, any action deterministically yields the intuitive outcome. With time, when applying Left or Right, the probability to reach the positions usually stemming from Up and Down increases symmetrically until reaching 0.45. We set the Lipschitz constant $L_p = 1$. Aside, we introduce a parameter $\epsilon \in [0, 1]$ controlling the behavior of the environment. If $\epsilon = 0$, only the left-hand side bridge becomes slippery with time. It reflects a close to worst-case evolution for a policy aiming to the left-hand side goal. If $\epsilon = 1$, only the right-hand side bridge becomes slippery with time. It

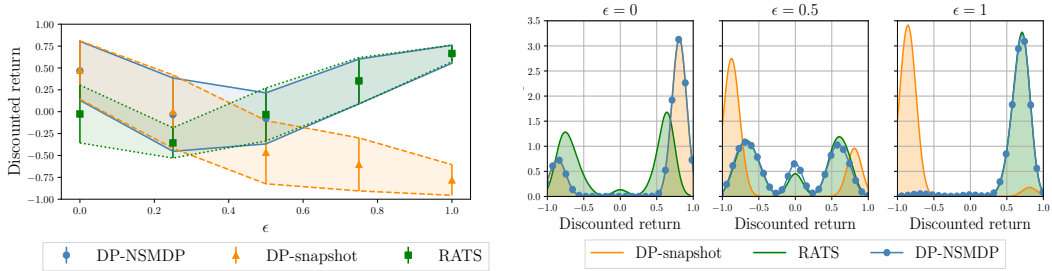

(a) Discounted return vs $\epsilon$, 50% of standard deviation.

(b) Discounted return distributions $\epsilon \in \{0, 0.5, 1\}$.

Figure 2: Discounted return of the three algorithms for various values of $\epsilon$.

reflects a close to worst-case evolution for a policy aiming to the right-hand side goal. In between, the misstep probability is proportionally balanced between left and right. One should note that changing $\epsilon$ from 0 to 1 does not cover all the possible evolutions from $\text{MDP}_{t_0}$ but provides a concrete, graphical illustration of RATS's behavior for various possible evolutions of the NSMDP.

We tested RATS with $d_{\max} = 6$ so that leaf nodes in the search tree are terminal states. Hence, the optimal risk-averse policy is applied and no heuristic approximation is made. Our goal is to demonstrate that planning in this worst-case NSMDP allows to minimize the loss given any possible evolution of the environment. To illustrate this, we report results reflecting different evolutions of the same NSMDP using the $\epsilon$ factor. It should be noted that, at $t = 0$, RATS always moves to the left, even if the goal is further, since going to the right may be risky if the probabilities to go Up and Down increase. This corresponds to the careful, risk-averse, behavior. Conversely, DP-snapshot always moves to the right since $\text{MDP}_0$ does not capture this risk. As a result, the $\epsilon = 0$ case reflects a favorable evolution for DP-snapshot and a bad one for RATS. The opposite occurs with $\epsilon = 1$ where the cautious behavior dominates over the risky one, and the in-between cases mitigate this effect.

In Figure 2a, we display the achieved expected return for each algorithm as a function of $\epsilon$, i.e. as a function of the possible evolutions of the NSMDP. As expected, the performance of DP-snapshot strongly depends on this evolution. It achieves high return for $\epsilon = 0$ and low return for $\epsilon = 1$. Conversely, the performance of RATS varies less across the different values of $\epsilon$. The effect illustrated here is that RATS maximizes the minimal possible return given *any* evolution of the NSMDP. It provides the guarantee to achieve the best return in the worst-case. This behavior is highly desirable when one requires robust performance guarantees as, for instance, in critical certification processes. Figure 2b displays the return distributions of the three algorithms for $\epsilon \in \{0, 0.5, 1\}$. The effect seen here is the tendency for RATS to diminish the left tail of the distribution corresponding to low returns for each evolution. It corresponds to the optimized criteria, i.e. robustly maximizing the worst-case value. A common risk measure is the Conditional Value at Risk (CVaR) defined as the expected return in the worst $q\%$ cases. We illustrate the CVaR at 5% achieved by each algorithm in Table 1b. Notice that RATS always maximizes the CVaR compared to both DP-snapshot and DP-NSMDP. Indeed, even if the latter uses the true model, the optimized criteria in DP is the expected return.

## 7  Conclusion

We proposed an approach for robust planning in non-stationary stochastic environments. We introduced the framework of Lipchitz Continuous Non-Stationary MDPs (NSMDPs) and derived the Risk-Averse Tree-Search (RATS) algorithm, to predict the worst-case evolution and to plan optimally w.r.t. this worst-case NSMDP. We analyzed RATS theoretically and showed that it approximates a worst-case NSMDP with a control parameter that is the depth of the search tree. We showed empirically the benefit of the approach that searches for the highest lower bound on the worst achievable score. RATS is robust to every possible evolution of the environment, i.e. maximizing the expected worst-case outcome on the whole set of possible NSMDPs. Our method was applied to the uncertainty on the evolution of a model. Generally, it could be extended to any uncertainty on the model used for planning, given bounds on the set of the feasible models. The purpose of this contribution is to lay a basis of worst-case analysis for robust solutions to NSMDPs. As is, RATS is computationally intensive and scaling the algorithm to larger problems is an exciting future challenge.

## Acknowledgments

This research was supported by the Occitanie region, France.

## Footnotes

[1]  Code: https://github.com/SuReLI/rats-experiments – ML reproducibility checklist: Appendix Section 8.

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
