[Supplementary Material]

# Non-Stationary Markov Decision Processes
# a Worst-Case Approach using Model-Based
# Reinforcement Learning

# Appendix

**Erwan Lecarpentier**
Université de Toulouse
ONERA - The French Aerospace Lab
erwan.lecarpentier@isae-supaero.fr

**Emmanuel Rachelson**
Université de Toulouse
ISAE-SUPAERO
emmanuel.rachelson@isae-supaero.fr

In the following proofs, the dual formulation of the 1-Wasserstein distance is used several times. We include the definition here for reference purpose.

**Definition 1.** *Dual formulation of the 1-Wasserstein distance. Let $(X, d_X)$ be a Polish metric space and $\mu, \nu$ any two probability measures on $X$. The dual formulation of the 1-Wasserstein distance between $\mu$ and $\nu$ is defined by*

$$W_1(\mu, \nu) = \sup_{f \in Lip_1} \int_X f(x) d(\mu - \nu)(x) \tag{1}$$

*where $Lip_1$ denotes the set of the continuous mappings $X \rightarrow \mathbb{R}$ with a minimal Lipschitz constant bounded by $1$.*

## 1 Proof of Property 1

Consider an $(L_p, L_r)$-LC-NSMDP. Let $s, t, a, \hat{t} \in \mathcal{S} \times \mathcal{T} \times \mathcal{A} \times \mathcal{T}$ be. By definition of the expected reward function, the following holds:

$$
\begin{aligned}
R_t(s, a) - R_{\hat{t}}(s, a) &= \int_{\mathcal{S}} \Big( p_t(s' \mid s, a) r_t(s, a, s') - p_{\hat{t}}(s' \mid s, a) r_{\hat{t}}(s, a, s') \Big) ds' \\
&= \int_{\mathcal{S}} \Big( r_t(s, a, s') \big[ p_t(s' \mid s, a) - p_{\hat{t}}(s' \mid s, a) \big] \\
&\quad + p_{\hat{t}}(s' \mid s, a) \big[ r_t(s, a, s') - r_{\hat{t}}(s, a, s') \big] \Big) ds' \\
&= \int_{\mathcal{S}} r_t(s, a, s') \big[ p_t(s' \mid s, a) - p_{\hat{t}}(s' \mid s, a) \big] ds' \\
&\quad + \int_{\mathcal{S}} p_{\hat{t}}(s' \mid s, a) \big[ r_t(s, a, s') - r_{\hat{t}}(s, a, s') \big] ds' \\
&\leq \sup_{\|f\|_L \leq 1} \int_{\mathcal{S}} f(s', t') \big[ p_t(s' \mid s, a) - p_{\hat{t}}(s' \mid s, a) \big] ds' \\
&\quad + \int_{\mathcal{S}} p_{\hat{t}}(s' \mid s, a) L_r |t - \hat{t}| ds' \\
&\leq W_1(p(\cdot \mid s, t, a), p(\cdot \mid s, \hat{t}, a)) + L_r |t - \hat{t}| \\
&\leq (L_p + L_r)|t - \hat{t}|
\end{aligned}
$$

Where we used the triangle inequality, the fact that $r$ is a bounded function and the dual formulation of the 1-Wasserstein distance (see Definition 1). The same inequality can be derived with the opposite terms which concludes the proof by taking the absolute value.

## 2 Proof of Property 2

*Proof.* The proof is straightforward using the Lipschitz property of Definition 4 and Property 1. $\square$

## 3 Proof of Property 4

Let us first calculate the cost of constructing a tree with the minimax procedure. Following Algorithm 1, a tree is composed of at most $n_l$ leaf nodes, $n_d$ non-leaf decision nodes and $n_c$ chance nodes, with the following values for the integers $n_l$, $n_d$ and $n_c$:

$$n_l = (|\mathcal{S}||\mathcal{A}|)^{d_{\max}}, n_d = \sum_{i=0}^{d_{\max}-1} (|\mathcal{S}||\mathcal{A}|)^i, \text{ and } n_c = |\mathcal{A}|B.$$

As a result, we have that $n_l$ is $\mathcal{O}((|\mathcal{S}||\mathcal{A}|)^{d_{\max}})$, $n_d$ is $\mathcal{O}((|\mathcal{S}||\mathcal{A}|)^{d_{\max}-1})$ and $n_c$ is $\mathcal{O}(|\mathcal{A}|(|\mathcal{S}||\mathcal{A}|)^{d_{\max}-1})$. We note respectively $c_l$, $c_d$ and $c_c$ the number of operations required to compute the values of a leaf node, a non-leaf decision node and a chance node. To compute the whole tree we need to build *and* evaluate all the nodes, resulting in at most the following number of operations:

$$n_l c_l \times n_d c_d \times n_c c_c. \tag{2}$$

We will assume that $c_l$ is $\mathcal{O}(1)$ without further details on the nature of the heuristic function. As the value of a non-leaf decision node is computed by finding the maximum value among the $|\mathcal{A}|$ children, we have that $c_d$ is $\mathcal{O}(|\mathcal{A}|)$. From Theorem 3, the evaluation of a chance node is equivalent to computing a 1-Wasserstein distance, which is a linear program. Following Vaidya's algorithm [Vaidya, 1989], the cost in the worst-case is $\mathcal{O}(|\mathcal{S}|^{2.5})$ where $|\mathcal{S}|$ is the dimension of the problem in our case. As a result, $c_c$ is $\mathcal{O}(|\mathcal{S}|^{2.5})$. Replacing all the values in Equation 2, we deduce that the total number of operation of computing a tree is

$$\mathcal{O}\left(|\mathcal{S}|^{1.5}(|\mathcal{S}||\mathcal{A}|)^{d_{\max}}\right).$$

After computing a tree, the action maximizing the value should be selected which has complexity $\mathcal{O}(|\mathcal{A}|)$. The operation being repeated for every time steps, one should multiply everything by $B$, the total number of time steps for which the algorithm is run. As a result, the total computational complexity of RATS is

$$\mathcal{O}\left(B|\mathcal{S}|^{1.5}|\mathcal{A}|(|\mathcal{S}||\mathcal{A}|)^{d_{\max}}\right).$$

## 4 Proof of Property 3

We are looking for a closed-form expression of the value of a chance node $\nu^{s,t,a}$ as defined in Equation 6 recalled below.

$$(\bar{p}, \bar{R}) = \arg\min_{(p,R)\in\Delta_{t_0,t}} R(s,a) + \gamma \mathbb{E}_{s'\sim p(\cdot|s,a)} V(\nu^{s',t+1})$$

Obviously, we have that $\bar{R} = R_{t_0}(s,a) - L_R|t - t_0|$ and $\bar{p}$ is given by:

$$\bar{p} = \arg\min_{p\in\mathcal{B}_{W_1}(p_{t_0}(\cdot|s,a),L_p|t-t_0|)} \sum_{s'} p(s' \mid s,a) V(\nu^{s',t+1})$$

where $\mathcal{B}_d(c,r)$ denotes the ball of center $c$, defined with metric $d$ and radius $r$. Since we are in the discrete case, we enumerate through the elements of $\mathcal{S}$ and write the vectors $p \equiv (p(s' \mid s,a))_{s'}$,

$p_0 \equiv (p_{t_0}(s' \mid s, a))_{s'}$ and $v \equiv (V(\nu^{s',t+1}))_{s'}$. The problem can then be re-written as follows:

$$\bar{p} = \arg\min_p \quad p^\top v \tag{3}$$

$$\text{s.t. } p^\top \mathbf{1} = 1 \tag{4}$$

$$p \geq 0 \tag{5}$$

$$W_1(p, p_0) \leq C \tag{6}$$

Where we have $\mathbf{1} \in \mathbb{R}^{|\mathcal{S}|}$ a vector of ones, $C = L_p|t - t_0|$ and the 1-Wasserstein metric between two discrete distributions written in dual form following Lemma 1 as:

$$W_1(u, v) = \max_f f^\top(u - v) \tag{7}$$

$$\text{s.t. } Af \leq b$$

Where the matrix $A$ and vector $b$ are defined such that for any indexes $i, j$ we have $|f_i - f_j| \leq d_{i,j}$ with $d_{i,j}$ the metric defined over the measured space, in our case the state space $\mathcal{S}$. Hence we propose to solve the program 3 under constraints 4 to 6. Let us first show that this problem is convex. Clearly, the objective function in Equation 3 is linear, hence convex, and the constraints 4 and 5 define a convex set. We prove that the 1-Wasserstein distance is convex in Lemma 1.

**Lemma 1.** *Convexity of the 1-Wasserstein distance. The 1-Wasserstein distance is convex i.e. for $\lambda \in [0, 1]$, $(X, d_X)$ a Polish space and any three probability measures $w_0, w_1, w_2$ on $X$, the following holds:*

$$W_1(w_0, \lambda w_1 + (1 - \lambda)w_2) \leq \lambda W_1(w_0, w_1) + (1 - \lambda)W_1(w_0, w_2)$$

*Proof.* We use the dual representation of the 1-Wasserstein distance of Definition 1.

$W_1(w_0, \lambda w_1 + (1 - \lambda)w_2)$

$$= \sup_{f \in \mathrm{Lip}_1} \int_X f(x)(w_0(x) - \lambda w_1(x) - (1 - \lambda)w_2(x))dx$$

$$= \sup_{f \in \mathrm{Lip}_1} \int_X (\lambda f(x)(w_0(x) - w_1(x)) + (1 - \lambda)f(x)(w_0(x) - w_2(x))) \, dx$$

$$\leq \lambda \sup_{f \in \mathrm{Lip}_1} \int_X f(x)(w_0(x) - w_1(x))dx + (1 - \lambda) \sup_{f \in \mathrm{Lip}_1} \int_X f(x)(w_0(x) - w_2(x))dx$$

$$\leq \lambda W_1(w_0, w_1) + (1 - \lambda)W_1(w_0, w_2)$$

Where we used the linearity of the integral and the triangle inequality on the sup operator. □

The program 3 is thus convex. One can also observe that the gradient of the objective function is constant, equal to $+v$. Furthermore, $p_0$ is an admissible initial point that we could use for a gradient descent method. However, given $p_0$, following the descent direction $-v$ may break the constraints 4 and 5. One would have to project this gradient onto a certain, unknown, set of hyperplanes in order to apply the gradient method descent. Let us note $\mathrm{proj}(v)$ the resulting projected gradient, that is unknown.

We remark that the vector $p_{\mathrm{sat}} = (0, \cdots, 0, 1, 0, \cdots, 0)$ with 1 at the index $\arg\min_i v_i$ where $v_i$ denotes the $i$th coefficient of $v$, is the optimal solution of the program 3 when we remove the Wasserstein constraint 6. One can observe that the optimal solution with the constraint 6 would as well be $p_{\mathrm{sat}}$ if the constant $C$ is *big enough*. As a result, the descent direction $\nabla = p_{\mathrm{sat}} - p_0$ is the one to be followed in this setting when applying the gradient descent method to this case. Furthermore, following $\nabla$ from $p_0$ until $p_{\mathrm{sat}}$ never breaks the constraints 4 and 5. Since the gradient of the objective function is constant, there can exist only one $\mathrm{proj}(v)$. $\nabla$ fulfils the requirements, hence we have $\mathrm{proj}(v) = \nabla$.

We can now apply the gradient method descent with the following 1-shot rule since the gradient is constant:

$$\bar{p} := p_0 + \lambda\nabla \text{ with, } \begin{cases} \lambda = 1 \text{ if } W_1(p_{\mathrm{sat}}, p_0) \leq C \\ \lambda = C/W_1(p_{\mathrm{sat}}, p_0) \end{cases}$$

Indeed, in the first case, we can follow $\nabla$ until the extreme distribution $p_{\text{sat}}$ without breaking the constraint 6. Going further is trivially infeasible.

In the second case, we have to stop *in between* so that the constraint 6 is saturated. In such a case, we cannot go further without breaking this constraint and we recall that no projected gradient could be found by uniqueness of this gradient in our setting. Hence we have the following equality:

$$W_1(p_0 + \lambda\nabla, p_0) = C$$
$$\max_{Af \leq b} f^\top (p_0 + \lambda\nabla - p_0) = C$$
$$\lambda \max_{Af \leq b} f^\top \nabla = C$$
$$\lambda = C/W_1(p_{\text{sat}}, p_0)$$

Where we used the fact that $\nabla = p_{\text{sat}} - p_0$. The latter result concludes the proof.

# 5 Proof of Property 5

Let us consider a tree developed with Algorithm 1 with a heuristic function $H : s \mapsto H(s)$ used to estimate the value of a leaf node. The set of the leaves nodes is denoted by $\mathcal{L}$ and we have the following uniform upper bound $\delta > 0$ on the heuristic error:

$$\forall \nu^{s,t} \in \mathcal{L}, \ |H(s) - \overline{V}^*_{t_0,t}(s)| < \delta \tag{8}$$

We want to prove the following result for a decision and chance nodes $\nu^{s,t}$ and $\nu^{s,t,a}$ at any depth $d \in [0, d_{\max}]$:

$$|V(\nu^{s,t}) - \overline{V}^*_{t_0,t}(s)| \leq \gamma^{(d_{\max}-d)}\delta \tag{9}$$
$$|V(\nu^{s,t,a}) - \overline{Q}^*_{t_0,t}(s,a)| \leq \gamma^{(d_{\max}-d)}\delta \tag{10}$$

The proof is made by induction, starting at depth $d_{\max}$ and reversely ending at depth 0. At $d_{\max}$, the nodes are leaf nodes, their values is estimated with the heuristic function i.e. $V(\nu^{s,t}) = H(s)$. Hence the result is directly proven by hypothesis in Equation 8. We will now start by proving the result for the chance nodes which come as the first parents of the decision node for which we initialized the induction proof. Then we extend it to the parents decision nodes which completes the proof.

**Chance nodes case.** Consider any chance node $\nu^{s,t,a}$ at depth $d \in [0, d_{\max}]$. We suppose the property is true for depth $d + 1$, thus we have for any decision node at $d + 1$ denoted by $\nu^{s',t'}$:

$$|V(\nu^{s',t'}) - \overline{V}^*_{t_0,t'}(s')| \leq \gamma^{(d_{\max}-(d+1))}\delta$$

Following Equation 6 of the paper, we have by construction:

$$V(\nu^{s,t,a}) = \overline{R}_t(s,a) + \gamma \sum_{s'} \overline{p}_t(s' \mid s, a)V(\nu^{s',t'})$$

By definition, the true $Q$-value function defined by the Bellman Equation 2 gives the true target value:

$$\overline{Q}^*_{t_0,t}(s,a) = \overline{R}_t(s,a) + \gamma \sum_{s'} \overline{p}_t(s' \mid s, a)\overline{V}^*_{t_0,t'}(s')$$

Hence, using the induction hypothesis, we have the following inequalities proving the result of Equation 10:

$$|V(\nu^{s,t,a}) - \overline{Q}^*_{t_0,t}(s,a)| = \gamma \left| \sum_{s'} \overline{p}_t(s' \mid s, a)V(\nu^{s',t'}) - \sum_{s'} \overline{p}_t(s' \mid s, a)\overline{V}^*_{t_0,t'}(s') \right|$$

$$\leq \gamma \sum_{s'} \overline{p}_t(s' \mid s, a) \left| V(\nu^{s'}) - \overline{V}^*_{t_0,t'}(s') \right|$$

$$\leq \gamma \sum_{s'} \overline{p}_t(s' \mid s, a)\gamma^{(d_{\max}-(d+1))}\delta$$

$$\leq \gamma^{(d_{\max}-d)}\delta$$

**Decision nodes case.** Consider now any decision node $\nu^{s,t}$ at the same depth $d \in [0, d_{\max})$. The value of such a node is given by Equation 5 of the paper and the following holds.

$$V(\nu^{s,t}) = V(\nu^{s,t,\bar{a}}), \text{ with, } \bar{a} = \arg\max_{a \in \mathcal{A}} V(\nu^{s,t,a})$$

Similarly, we define $a^* \in \mathcal{A}$ as follows:

$$\overline{V}^*_{t_0,t}(s) = \overline{Q}^*_{t_0,t}(s, a^*), \text{ with, } a^* = \arg\max_{a \in \mathcal{A}} \overline{Q}^*_{t_0,t}(s, a)$$

We distinguish two cases: 1) if $\bar{a} = a^*$ and 2) if $\bar{a} \neq a^*$. In case 1), the result is trivial by writing the value of the decision node as the value of the chance node with the action $a^*$ and using the – already proven for depth $d$ – result of Equation 10.

$$|V(\nu^{s,t}) - \overline{V}^*_{t_0,t}(s)| = |V(\nu^{s,t,a^*}) - \overline{Q}^*_{t_0,t}(s, a^*)|$$
$$\leq \gamma^{(d_{\max}-d)}\delta$$

In case 2), the maximizing actions are different. Still following Equation 10, we have that $V(\nu^{s,t,a^*}) \geq \overline{Q}^*_{t_0,t}(s, a^*) - \gamma^{(d_{\max}-d)}\delta$. Yet, since $\bar{a}$ is the maximizing action in the tree, we have that $V(\nu^{s,t,\bar{a}}) \geq V(\nu^{s,t,a^*})$. By transitivity, we can thus write the following:

$$V(\nu^{s,t,\bar{a}}) \geq \overline{Q}^*_{t_0,t}(s, a^*) - \gamma^{(d_{\max}-d)}\delta$$
$$\Rightarrow \quad \overline{Q}^*_{t_0,t}(s, a^*) - V(\nu^{s,t,\bar{a}}) \leq \gamma^{(d_{\max}-d)}\delta \tag{11}$$

Furthermore, still following Equation 10, we have that $\overline{Q}^*_{t_0,t}(s, \bar{a}) \geq V(\nu^{s,t,\bar{a}}) - \gamma^{(d_{\max}-d)}\delta$. Yet, since $a^*$ is the maximizing action in $\widehat{\mathrm{MDP}}$, we have that $\overline{Q}^*_{t_0,t}(s, a^*) \geq \overline{Q}^*_{t_0,t}(s, \bar{a})$. By transitivity, we can thus write the following:

$$\overline{Q}^*_{t_0,t}(s, a^*) \geq V(\nu^{s,t,\bar{a}}) - \gamma^{(d_{\max}-d)}\delta$$
$$\Rightarrow \quad V(\nu^{s,t,\bar{a}}) - \overline{Q}^*_{t_0,t}(s, a^*) \leq \gamma^{(d_{\max}-d)}\delta \tag{12}$$

By assembling equations 11 and 12, we prove equation 9 and the proof by induction is complete.

## 6  Proof of Property 6

Let $s, t_0, t \in \mathcal{S} \times \mathcal{T} \times \mathcal{T}$ be. We consider the two snapshots $\mathrm{MDP}_{t_0}$ and $\mathrm{MDP}_t$ and are interested in the values of $s$ within those two snapshots using the random policy $\pi$. We note $V^\pi_{\mathrm{MDP}_{t_0}}(s)$ and $V^\pi_{\mathrm{MDP}_t}(s)$ those values. Let $n \in \mathbb{N}$ be. We note $V^{\pi,n}_{\mathrm{MDP}_{t_0}}(s)$ and $V^{\pi,n}_{\mathrm{MDP}_t}(s)$ the finite horizon values defined as follows:

$$V^{\pi,n}_{\mathrm{MDP}_{t_0}}(s) = \mathbb{E}\left\{\sum_{i=0}^n \gamma^i r_{t_0}(s_i, a_i, s_{i+i}) \,\middle|\, \begin{array}{l} s_0 = s, \\ s_{i+1} \sim p_{t_0}(\cdot \mid s_i, a_i),\ i \geq 0 \\ a_i \sim \pi(\cdot),\ i \geq 0 \end{array}\right\}$$

where we replace $t_0$ by $t$ for the definition of $V^{\pi,n}_{\mathrm{MDP}_t}(s)$. We first prove a result on the finite horizon values in Lemma 2.

**Lemma 2.** *We consider an $(L_p, L_R)$-LC-NSMDP. For $s, t, t_0 \in \mathcal{S} \times \mathcal{T} \times \mathcal{T}$ and $n \in \mathbb{N}$, the finite horizon of the values of $s$ within the snapshots $MDP_t$ and $MDP_{t_0}$ verify:*

$$|V^{\pi,n}_{MDP_{t_0}}(s) - V^{\pi,n}_{MDP_t}(s)| \leq L_{V_n}|t - t_0|$$

$$\text{with, } L_{V_n} = \sum_{i=0}^n \gamma^i L_R$$

*Proof.* The proof is made by induction. Let us start with $n = 0$. By definition, we have:

$$\left|V^{\pi,0}_{\mathrm{MDP}_{t_0}}(s) - V^{\pi,0}_{\mathrm{MDP}_t}(s)\right| = \left|\int_{\mathcal{A}} \pi(a \mid s)\left(R_{t_0}(s, a) - R_t(s, a)\right)da\right|$$
$$\leq \int_{\mathcal{A}} \pi(a \mid s)L_R|t_0 - t|da$$
$$\leq L_R|t_0 - t|$$

Which verifies the property for $n = 0$ with $L_{V_0} = L_R$. Let us now consider $n \in \mathbb{N}$ and suppose the property true for rank $n - 1$. By writing the Bellman equation for the two value functions, we obtain the following calculation:

$$V_{\mathrm{MDP}_{t_0}}^{\pi,n}(s) - V_{\mathrm{MDP}_t}^{\pi,n}(s) = \int_{\mathcal{S} \times \mathcal{A}} \pi(a|s) \Big[ p_{t_0}(s' \mid s, a)(r_{t_0}(s, a, s') + \gamma V_{\mathrm{MDP}_{t_0}}^{\pi,n-1}(s')) -$$

$$p_t(s' \mid s, a)(r_t(s, a, s') + \gamma V_{\mathrm{MDP}_t}^{\pi,n-1}(s')) \Big] ds' da$$

i.e. $V_{\mathrm{MDP}_{t_0}}^{\pi,n}(s) - V_{\mathrm{MDP}_t}^{\pi,n}(s) = \int_{\mathcal{A}} \pi(a|s) \Big[ A(s,a) + B(s,a) \Big] da$ \hfill (13)

With the following values for $A(s,a)$ and $B(s,a)$:

$$A(s,a) = \int_{\mathcal{S}} (r_{t_0}(s, a, s') + \gamma V_{\mathrm{MDP}_{t_0}}^{\pi,n-1}(s')) \Big[ p_{t_0}(s' \mid s, a) - p_t(s' \mid s, a) \Big] ds'$$

$$B(s,a) = \int_{\mathcal{S}} p_t(s' \mid s, a) \Big[ r_{t_0}(s, a, s') - r_t(s, a, s') + \gamma(V_{\mathrm{MDP}_{t_0}}^{\pi,n-1}(s') - V_{\mathrm{MDP}_t}^{\pi,n-1}(s')) \Big] ds'$$

Let us first bound $A(s,a)$ by noticing that $s' \mapsto r_{t_0}(s, a, s') + \gamma V_{\mathrm{MDP}_{t_0}}^{\pi,n-1}(s')$ is bounded by $\frac{1}{1-\gamma}$. Since the function $s' \mapsto \frac{1}{1-\gamma}$ belongs to $\mathrm{Lip}_1$, we can write the following:

$$A(s,a) \leq \sup_{f \in \mathrm{Lip}_1} \int_{\mathcal{S}} f(s') \Big[ p_{t_0}(s' \mid s, a) - p_t(s' \mid s, a) \Big] ds'$$

$$\leq W_1(p_{t_0}, p_t)$$

$$\leq L_p |t - t_0|$$

$B$ is straightforwardly bounded using the induction hypothesis:

$$B(s,a) \leq \int_{\mathcal{S}} p_t(s' \mid s, a) \Big[ L_r |t - t_0| + \gamma \sum_{i=0}^{n-1} \gamma^i L_R |t - t_0| \Big] ds'$$

$$\leq L_r |t - t_0| + \sum_{i=1}^{n} \gamma^i L_R |t - t_0|$$

We inject the result in Equation 13:

$$V_{\mathrm{MDP}_{t_0}}^{\pi,n}(s) - V_{\mathrm{MDP}_t}^{\pi,n}(s) \leq \int_{\mathcal{A}} \pi(a|s) \Big[ L_p |t - t_0| + L_r |t - t_0| + \sum_{i=1}^{n} \gamma^i L_R |t - t_0| \Big] da$$

$$\leq (L_p + L_r)|t - t_0| + \sum_{i=1}^{n} \gamma^i L_R |t - t_0|$$

$$\leq L_R |t - t_0| + \sum_{i=1}^{n} \gamma^i L_R |t - t_0|$$

$$\leq \sum_{i=0}^{n} \gamma^i L_R |t - t_0|$$

The same result can be derived with the opposite expression. Hence, taking the absolute value, we prove the property at rank $n$, i.e.

$$|V_{\mathrm{MDP}_{t_0}}^{\pi,n}(s) - V_{\mathrm{MDP}_t}^{\pi,n}(s)| \leq \sum_{i=0}^{n} \gamma^i L_R |t - t_0| \hfill (14)$$

which concludes the proof by induction. $\hfill \square$

The proof of Property 6 follows easily by remarking that the sequence $L_{V_n}$ of Lemma 2 is geometric and converges towards $\frac{L_R}{1-\gamma}$ when $n$ goes to infinity.

Figure 1: The Non-Stationary bridge environment

## 7 Non-Stationary bridge environment

## 8 Informations about the Machine Learning reproducibility checklist

For the experiments run in Section 6, the computing infrastructure used was a laptop using four 64-bit CPU (model: Intel(R) Core(TM) i7-4810MQ CPU @ 2.80GHz). The collected samples sizes and number of evaluation runs for each experiment are summarized in Table 1.

| Experiment | Number of experiment repetitions | Number of episodes | Maximum length of episodes | Upper bound on the number of computed transition samples $(s, a, r, s')$ |
|---|---|---|---|---|
| Non-Stationary Bridge Figure 1 | 3 (one per agent) | 96 | 10 | 89,579,520 |

Table 1: Summary of the number of experiment repetition, number of sampled tasks, number of episodes, maximum length of episodes and upper bounds on the number of collected samples.

The displayed confidence intervals in Figure 2a is 50% of the estimated confidence interval $\bar{\sigma}$ computed w.r.t. the following formula:

$$\bar{\sigma} = \sqrt{\frac{1}{1-N} \sum_{i=1}^{N} (x_i - \bar{x})^2} \quad \text{where,} \quad \bar{x} = \frac{1}{N} \sum_{i=1}^{N} x_i,$$

with $D = \{x_i\}_{i=1}^{N}$ the set of the collected data (discounted return in this case). No data were excluded neither pre-computed. Hyper-parameters were determined to our appreciation, they may be sub-optimal but we found the results convincing enough to display interesting behaviours.

## References

Pravin M. Vaidya. Speeding-up linear programming using fast matrix multiplication. In *30th Annual Symposium on Foundations of Computer Science*, pages 332–337. IEEE, 1989.