[Reviews · NeurIPS 2019]

Reviewer 1



UPDATE: I have read the authors response and increased my score. Specifically, the authors fixed my understanding of Property 1 and properly framed the relaxation of the problem in Section 5. Please include similar clarifications in the final work. There was also a lot of discussion among the reviewers about how the paper relates to the Robust MDP literature, which needs to be covered better in the current work. Papers such as "Reinforcement Learning in Robust Markov Decision Processes" and "Online Learning in Markov Decision Processes with Adversarially Chosen Transition Probability Distributions" were brought up by others and seem applicable in the current setting and could be empirical competitors to RATS. I very much like the constraints used to study planning in non-stationary environments in this paper and the min-max inspired RATS algorithm seems like an appropriate game theoretic approach. However, there are several clarity points that need to be addressed in the assumptions, and I feel the experiments need at least one other competitor from the long lineage of non-stationary planning or game theoretic decision-making approaches. Still the paper as is seems correct and makes progress in the literature. Clarifications needed: Property 1 seems stronger than I would have expected. By bounding changes in the reward function based on evolution of the transition function, you are assuming their evolution is correlated, and that lays the framework for the ultimate proof because the resulting Bellman backup will be bounded. Can the authors comment on how often they expect Property 1 to hold in practice? Are there environments they encountered where it did or did not hold. To my reading it sounds like a very strong assumption. The description of the assumptions around line 205 need clarification. I find this section difficult to follow. The authors claim be solving a relaxation of the problem, and violating property 2, but it is unclear to me what the new assumptions are. Can the authors clearly state “considering snapshots only constrained by MDP_0” actually means? Property 4 needs justification in the paper itself. Particularly, the construction and solving of the linear program referenced in the supplemental material should be mentioned here. It is an important detail. The experiments are both informative, but ultimately seem incomplete. I actually appreciate having a small grid-world where one can easily analyze the behavior of the algorithm, and I liked having the two book-ends as the competitors. I learned something from this experiment. However, I was disappointed that (1) none of the many other methods mentioned in the related work were applied, even if their assumptions did not hold, to see if RATS is really state of the art and (2) after all of the discussion of a heuristic function, there was no experimentation with various heuristics. Lesser points: Definition 3 – While Wasserstein is an acceptable choice in continuous physical environments, it seems like total variation may be a better fit in discrete environments. Indeed, much sample complexity analysis in discrete stationary MDPs utilize TV to compare learned models, so I would urge the authors to note this may yet be an acceptable measure in those environments, even though the current work focuses on more continuous domains. The literature review is fairly complete but lack discussion of game-theoretic tree search approaches, which seem highly applicable (though ultimately probably overpowered) here. Line 90 consider -> considers

Reviewer 2



Summary: This paper addresses the problem of non-stationary MDPs where the MDP continuously and smoothly changes over time even though this evolution is unknown (only the current MDP is known). More precisely, the non-stationary transition probability p_t is assumed to be Lipschitz-continuous in t w.r.t. the Wasserstein distance and the non-stationary reward is also assumed to be Lipschtz-continuous in t. To solve this problem, a new algorithm called RATS (Risk Averse Tree Search) is introduced. RATS assumes a worst-case evolution of the MDP and implements a minimax procedure. RATS is empirically evaluated on a toy domain. It performs similarly to an oracle that explicitly knows the evolution of p_t and r_t and clearly outperforms dynamic programming approaches that do not take into account non-stationarities. Clarity: The paper is well-written and all assumptions and claims are properly justified. The proofs are easy to follow and the algorithm is well-explained. Correctness: I only noticed a small typo in the proof of property 3: between lines 64 and 65, I think \lambda = C/ W_1(...) (the / is missing). This typo is also present in property 3 where I think that \lambda = L_p |t-t_0| / W_1(...) which seems more plausible. I think everything else is correct. Significance: The setting considered in this paper is new and relevant. The algorithm is also new and solves the problem. The authors acknowledge the weaknesses of their approach (e.g., lack of scalability of the algorithm). The paper may be borderline in terms of contributions for a conference like NeurIPS but I think overall it is good enough. When you mention the case where indices do not coincide with time (lines97 & 98), do you have in minds Semi-Markov Decision Processes? It would be worth mentionning the connection between the set of admissible snapshots and bounded-parameter MDPs ("Bounded Parameter Markov Decision Processes", R. Givan, S. Leach and T. Dean, 1997).

Reviewer 3



The paper proposes a problem of planning in non-stationary environments where the change is gradual. The gradual change assumption is formalized by stating a Lipschitzness assumption. The authors assume that only the model evolution is not known but the agent has access to the current model. Given this along with the Lipschitz assumption, the method constructs a set of admissible MDPs. A worst case value function is defined using this admissible set which can be easily derived from the smoothness property. For worst case planning with this set, the problem becomes combinatorial in specification as the constraint has to be respected for all time-steps. To avoid this, the authors relax the problem to a case where the constraints are only verified for with respect to the planning timestep t_0. With this, a minimax tree search approach is proposed and analyzed. The proofs are quite simple to understand and to my knowledge have no errors. However, the algorithm description can be a bit more clearer with apt description of the bottom up minimax planning description. A few comments regarding the methods: 1. Since, the Lipschitz assumption is taken over the time indices, the constant has to be quite small for the admissible models to be non-vacuous. Can the authors comment on the plausibility of this assumption? How does it compare with a setting where there is a bounded change assumption. What modification would the authors make for such a setting to RATS? 2. Since, the method is not computationally scalable, it is desirable to have a discussion of where things can be changed. 3. Can any guarantee be given for the relation between the actual objective (1,2) vs the relaxed version (3,4)? The related work in non-stationary learning problems is very nicely compiled. However, given the risk-aware planning nature of the problem, I think there should be a discussion in related work and experiments by considering methods from the robust MDP literature. The setting is arguably an instance of that where the robust (maxmin) planning needs to be done for the plausible MDP set. Also, the authors state this to be a zero-shot planning. It would be helpful to clearly state what the authors mean by zero-shot exactly in this setting. Further, apart from the theoretical contributions, a discussion about where this problem is relevant, practical instances and possibly a set of such experiments would strengthen the paper. ------------------------------- Update: I have read the author response and would maintain my initial rating. The technical contribution here is limited as it is not significantly challenging to apply minimax tree search algorithm to the rectangular uncertainty set after the relaxation. It would be nice to see how the relaxation affects the outcome by comparing with the brute force result in small domains.

[Author Response · NeurIPS 2019]

We thank the reviewers for their thoughtful reviews and comments. We intend to include the answers below in the paper.

**Clarifying statements.** Thank you (reviewer #4) for pointing out these weaknesses that can easily be fixed. We believe we attempted to lay bridges between various fields (tree search, robust MDPs, non-stationary planning...) which could explain the difference in clarity appreciation between reviewers. We elaborate on specific clarifications below.

**When does Property 1 hold?** There may be a misunderstanding here; Property 1 is not an assumption. It is a direct consequence of the LC-NSMDP definition and thus always holds.

**Section 5's relaxation vs. full problem.** In the full problem (Equation 1), $MDP_t$ belongs to a set of MDPs defined (constrained) by $MDP_{t-1}$. This is a stronger requirement than the one used by RATS, which only constrains $MDP_t$ to belong to the "cone" of MDPs originating from $MDP_{t_0}$ (see figure). There currently is no generic bound on the optimality gap between the relaxation and the full problem. Although this is an interesting problem, this relaxation was only introduced to allow the bottom-up minimax method and we believe further research should rather investigate alternative robust algorithms that lift this relaxation.

**Wasserstein metric (WM) vs. total variation (TV).** Indeed, TV is a legitimate measure in *non-metric* state spaces. However, many discrete state spaces (as the ones used in the experiments) still exhibit a metric between states and WM computes distances that depend on that metric while TV does not. Not taking that metric into account would yield worst case snapshots with little physical meaning (think of a path planning task where the worst case snapshot transforms the outcome distribution of a "turn right" transition from a street in Montreal to a street in Vancouver).

**On the experimental section.** The purpose of this work was really to lay ground for a principled approach to NSMDP planning. We believe the main contribution of this section is the comparison between robust and non-robust policies on a variety of demonstrative scenarios. Although it is an important area of future work, the goal to scale up seemed like a drift from this main goal. On that topic, we argue that RATS aims at the same result as Robust Dynamic Programming (RDP, Iyengar 2005) and, for the benchmarks reported, yields the same policy. Furthermore, the comparison with other related approaches (RDP seeming the most appropriate, because non-stationary planning assumes full model knowledge) would be relevant in a context where one wants to scale to larger problems, including those where function approximation is needed. In particular, such a comparison should highlight that RDP does DP —it is offline and plagued by the curse of dimensionality— while RATS does tree search from the current state and snapshot —it is online and can take advantage of heuristic search. Scaling up, as in many game-theoretic approaches, likely requires a combination of online search, function approximation and relevant heuristics, and we believe this topic deserves another paper. We agree nonetheless that the question comes naturally and intend to add the present paragraph to the experimental section.

**Extension to continuous time processes.** The extension of our work to *deterministic* continuous durations is straightforward. We chose to present our work in the case where decision epochs and epoch indices coincide for clarity but will include an additional paragraph in the appendix to lift this ambiguity. Considering *stochastic* continuous durations can be done in a number of ways. Semi-MDPs (and their extensions, *e.g.* generalized SMDPs) assume such durations but in a time-independent, stationary setting. Having both time-dependency (non-stationarity) and stochastic continuous durations introduces an additional layer of complexity and the extension of our results in that case might be feasible but seems non-trivial.

**Bounded parameters MDPs.** We omitted the reference because it is a precursor of the Robust MDP literature, but we definitely acknowledge the link with the $\Delta_t$ set (and the rectangularity assumption in RDP) so will try to include it back.

**Bounded evolution vs. bounded change rates.** Bounding the absolute evolution is an interesting extension that sits between our model and Robust MDPs. We could include it in Equation 6 by adding a *constant* upper bound on the distance to $MDP_{t_0}$. Assuming bounded evolution amounts to say "the model cannot evolve far from the current snapshot" while bounded change rate says "the model can evolve arbitrarily far from the current snapshot but slowly". In practice, both assumptions make sense for real world instances. Thank you for this very relevant comment, this will be a valuable addition in the paper.

**Zero-shot planning.** We agree this terminology is confusing (we meant "without data acquisition" as often in zero-shot learning) and intend to remove it altogether.

**Practical problem instances.** This research was inspired by path planning problems in autonomous glider soaring and road traffic management. These examples are mentioned in Sections 2 and 4 but could be present since Section 1.

[Meta-Review · NeurIPS 2019]

The reviewers felt that this paper was well-executed, even though the proposed approach is a rather straightforward application of techniques from the robust MDP literature (specifically, minmax planning with appropriately defined uncertainty sets derived from a Lipschitzness assumption). For the final version, the authors should improve the discussion of related literature on robust MDPs (e.g., "Reinforcement Learning in Robust Markov Decision Processes" by Lim et al., NIPS 2013 + references therein) and on MDPs with non-stationary transitions (e.g., "Online Learning in Markov Decision Processes with Adversarially Chosen Transition Probability Distributions" by Abbasi-Yadkori et al., NIPS 2013 + references therein).